# Body Mass Index Is Independently Associated with the Presence of Ischemia in Myocardial Perfusion Imaging

**DOI:** 10.3390/medicina58080987

**Published:** 2022-07-25

**Authors:** Chrissa Sioka, Paraskevi Zotou, Michail I. Papafaklis, Aris Bechlioulis, Konstantinos Sakellariou, Aidonis Rammos, Evangelia Gkika, Lampros Lakkas, Sotiria Alexiou, Pavlos Kekiopoulos, Katerina K. Naka, Christos Katsouras

**Affiliations:** 1Department of Nuclear Medicine, University Hospital of Ioannina, 45500 Ioannina, Greece; pzotoy1966@gmail.com (P.Z.); sakellarius@yahoo.gr (K.S.); evagkika@yahoo.gr (E.G.); sotiria.alexiou@yahoo.com (S.A.); pavlos_kek@hotmail.com (P.K.); 22nd Department of Cardiology, University Hospital of Ioannina, 45500 Ioannina, Greece; m.papafaklis@yahoo.com (M.I.P.); md02798@yahoo.gr (A.B.); aidrammos@yahoo.gr (A.R.); ftpcavalier52@gmail.com (L.L.); anaka@uoi.gr (K.K.N.); cskats@yahoo.com (C.K.)

**Keywords:** myocardial perfusion imaging, myocardial ischemia, body mass index, waist circumference, obesity

## Abstract

*Background and Objectives*: Obesity has been linked to various cardiovascular risk factors, increased incidence of coronary artery disease, and myocardial perfusion defects. The aim of this study was to investigate if body mass index (BMI) and waist circumference (WC) were associated with myocardial perfusion defects. *Materials and Methods*: A total of 308 consecutive patients who had myocardial perfusion imaging (MPI) with single photon emission computed tomography (SPECT) and a complete medical record on file were studied retrospectively. *Results*: The median age was 69 (61–76) years, the BMI was 27.6 (24.4–30.7) kg/m^2^, and the WC was 110 (102–118) cm. Of the 308 patients, 239 patients (77.6%) had myocardial ischemia. A positive test for ischemia was more frequent in men compared to women (72 vs. 28%, *p* < 0.001). Within the male group, BMI and WC were not significantly different between the ischemia and non-ischemia groups. In contrast, within the female group, both BMI (30.2 vs. 27.1 kg/m^2^, *p* = 0.002) and WC (112 vs. 105.5 cm, *p* = 0.020) were significantly higher in the ischemia group. Multivariable logistic regression showed that male sex and BMI were the only two independent predictors of ischemia in our patient population. *Conclusions*: This study showed that BMI was an independent predictor of ischemia in our patient population.

## 1. Introduction

An imaging method depicting perfusion of the cardiac muscle that is low cost and widely available is myocardial perfusion imaging (MPI) with single photon emission computed tomography (SPECT). This technology is vital for the assessment of the extent of myocardial ischemia due to coronary artery disease (CAD), while it can monitor the response to various therapeutic interventions and patient prognosis [1,2,3,4]. MPI may provide important information about the existence and severity of myocardial ischemia not only in patients with cardiovascular diseases but also in several other diseases and especially in obese individuals who are not appropriate candidates for other imaging methods [5]. The diagnostic accuracy of MPI appears comparable to coronary angiography for both sexes, with a sensitivity of 84.2% and 89.1% and a specificity of 78.7% and 71.2% for females and males, respectively [6]. Moreover, MPI may provide important information for myocardial ischemia not related to CAD but to other rare causes [7,8,9,10,11]. One such medical condition is ischemia without obstructive stenosis, frequently linked to coronary microvascular dysfunction [10,11,12,13].

Obesity is a pandemic of the modern era and has been related to adverse prognosis in various populations [14]. Cardiovascular diseases are a major cause of morbidity and mortality in obese patients and have been associated with the increased co-incidence of other classical cardiovascular risk factors in these patients (i.e., hypertension, diabetes mellitus, dyslipidemia, etc.) but also with obesity-specific pathophysiology leading to arterial atherosclerosis or cardiomyopathy [15]. Increased oxidative stress, inflammation, and insulin resistance (components of the metabolic syndrome) are some of the most important underlying abnormalities in obese patients linked to cardiovascular diseases [16,17,18,19]. However, the presence of various clinical phenotypes of obese patients with distinct prognosis, i.e., metabolically healthy vs. unhealthy obese, central vs. peripheral obesity, as well as the “obesity paradox” (i.e., less vascular atherosclerosis or improved prognosis in overweight and obese patients of specific populations) raise a debate about the actual link of obesity with cardiovascular diseases [20,21,22,23,24,25,26,27].

The aim of the present study was to assess the association of BMI and waist circumference (WC), an index of central obesity, with the presence of myocardial ischemia in patients undergoing MPI SPECT.

## 2. Materials and Methods

### 2.1. Subjects

We initially screened 314 consecutive patients who underwent an MPI study during 2016. However, 6 patients had to be excluded from the study due to the lack of adequate medical information on file (unclear history of known CAD or myocardial infarction). Thus, a total of 308 patients were included in the study.

All the body measurements were performed by trained staff. Height was measured to the nearest 0.1 cm using a stadiometer, and weight was measured to the nearest 0.1 kg by an electronic scale. Body mass index (BMI) was calculated as weight (kg) divided by the square of height (m). Waist circumference was measured to the nearest 0.1 cm using a non-stretchable tape measure over light clothing [28].

The study was performed after approval by the Hospital’s Clinical Research Committee that consented to the use of the anonymous clinical data of patients who participated. All medical data and the MPI results of patients were anonymized to be used in the analysis.

### 2.2. Scintigraphy

All patients were subjected to a 1-day MPI protocol in accordance with the published guidelines [29]. During the procedure, stress was accomplished either via the Bruce protocol treadmill exercise or with pharmacological exercise. During pharmacological or treadmill exercise, a reduction in oxygen supply to the myocardium induces ischemic changes that may be seen on a concomitant electrocardiogram (ECG) but can be assessed more accurately as a specific perfusion reduction on an MPI. Moreover, decreased uptake of the radiotracer to a specific region indicates the specific artery responsible for the myocardial ischemia of the patient. The difference in the radiotracer uptake at rest from the uptake during stress depicts the degree of ischemia reversibility [1].

MPI images were visually evaluated utilizing a polar myocardium map divided into 17 segments, each one scored on a 5-point severity scale. In each of the 17 segments, the uptake was considered normal when scored with 0, as mildly reduced uptake with score 1, moderately reduced uptake with score 2, severely reduced uptake with score 4, and without uptake with score 5. The summed scores were calculated during both stress (SSS) and rest (SRS). The difference between the SSS and SRS was also assessed, demonstrating the degree of reversible perfusion defects (reversible ischemia). An MPI exam was defined as normal when the SSS was less than 4 and abnormal when the SSS was equal to or over 4 [30,31,32].

### 2.3. Statistical Analysis

Categorical variables were presented as counts and percentages and were compared using the chi-square or Fisher’s exact test as appropriate. The normal distribution of continuous data was tested using the Kolmogorov–Smirnov test, and the variables of interest (i.e., age, BMI, and WC) were found to deviate from the normal distribution. Accordingly, continuous variables were presented as medians (interquartile range (IQR)). Nonparametric analysis using the Wilcoxon–Mann–Whitney test was employed to compare continuous variables between the two groups (ischemia and non-ischemia). Logistic regression analysis was performed to investigate the association of baseline variables with the presence of ischemia in myocardial scintigraphy. The variables associated with ischemia in univariate analysis at *p* level ≤0.2 were entered into a multivariable logistic regression model with a stepping algorithm to identify the independent predictors of ischemia. The level of significance was set at *p* < 0.05, and the statistical tests were two-tailed. The IBM SPSS statistical software package (version 26 for Windows, Armonk, NY, USA) was used for the analysis.

## 3. Results

In the total population, median (IQR) age, BMI, and WC were 69 (61–76) years, 27.6 (24.4–30.7) kg/m^2^, and 110 (102–118) cm, respectively. Evidence of ischemia in myocardial scintigraphy was found in 239 patients (77.6%). Although the age was not different between the ischemia and non-ischemia groups, the median BMI was numerically higher without reaching statistical significance (*p* = 0.069), and the WC was significantly larger (*p* = 0.039) in the ischemia group (Table 1). A positive test for ischemia was much more frequent in males compared to females (72 vs. 28%, *p* < 0.001), as well as in patients with a known history of CAD (*p* < 0.001), a history of previous myocardial infarction (*p* = 0.006), and a history of a previous coronary intervention (*p* = 0.008).

In subgroup analysis (primary grouping according to ischemia status; Figure 1), the BMI was significantly higher in females compared to males (30.1 vs. 26.8 kg/m^2^, *p* < 0.001) within the ischemia group, but it did not differ significantly between sexes within the non-ischemia group (27.1 vs. 24.8 kg/m^2^, *p* = 0.234). There was no significant difference regarding WC between males and females in either group. In further subgroup analysis (primary grouping according to sex; Figure 2), within the male group, BMI and WC were not significantly different between the two ischemia status groups. In contrast, within the female group, both BMI (30.2 vs. 27.1 kg/m^2^, *p* = 0.002) and WC (112 vs. 105.5 cm, *p* = 0.020) were significantly higher in the ischemia group.

In univariate analysis for the prediction of the presence of ischemia in MPI (Table 2), the male sex had the strongest association with ischemia (odds ratio: 6.76, 95% CI: 3.71–12.3, *p* < 0.001), while a history of known CAD had the second strongest association (odds ratio: 6.76, 95% CI: 3.71–12.3, *p* < 0.001). The multivariable logistic regression showed that male sex and BMI were the only two independent predictors of ischemia in our patient population (Table 3).

## 4. Discussion

Obesity, a putative cardiovascular risk factor, is generally increasing in most countries [33]. BMI and WC are indices of obesity and are considered to provide independent information about the outcome in patients with CAD [34].

In the present study, the univariate analysis depicted a statistically significant association between higher WC and myocardial ischemia (*p* = 0.028) and a trend towards a higher BMI in the myocardial ischemia group (*p* = 0.062). However, the multivariable logistic regression showed that BMI was an independent predictor of ischemia in our patient cohort. Our results are in accordance with a previous study by Demirci et al. [35], that evaluated the role of obesity (overweight, grade I obesity, and severe obesity) in 1005 patients who had a first episode of acute coronary syndrome. These authors reported that compared to normal weight patients, the first coronary episode occurred 3.9 years earlier in overweight patients, 6.1 years earlier in patients with grade I obesity, and 7.7 years earlier in patients with severe obesity.

Cardiovascular complications such as the development of CAD in obese patients are probably multifactorial. For example, obese individuals may develop type II diabetes mellitus due to insulin resistance [36,37], followed later by adverse lipid profiles, eventually leading to CAD or other severe cardiovascular complications [38,39]. However, even though obese individuals are more prone to CAD, there have been reports of better prognosis in patients with high BMI after coronary artery occlusion or acute myocardial infarction [40,41]. Underweight individuals (BMI < 18.5 kg/m^2^) with acute coronary syndromes exhibited a paradoxically worse prognosis than overweight individuals. These controversial results about the role of obesity in cardiovascular diseases have supported the obesity paradox [27]. In a study by Tsai et al., in 1301 patients with coronary artery occlusion, it was reported that patients with a high BMI had better outcome and lower all-cause mortality [41]. To explain this paradox of worse outcome in patients with lower BMI, some investigators have implicated progression of atherosclerosis of culprit lesions in vessels with small lumen diameters after PCI in these patients, even though other metabolic risk factors were absent [42]. Another longitudinal study of 4.4 years in 16,095 cases of myocardial infarction, 18,957 cases of stroke, and 30,200 cases of all-cause deaths demonstrated that increased variability of body weight and WC was associated with high risk for all three outcomes. These findings suggested that a variable, instead of a relatively stable body weight, either high or low, was associated with adverse prognosis [43]. A post-hoc breakdown of the SYNTAX Extended Survival study in patients with left main coronary artery disease that was revascularized either with coronary artery bypass surgery or percutaneous coronary intervention, showed that patients with either low baseline BMI and low WC, or low BMI/high WC, or high BMI/high WC, exhibited higher 10-year mortality risk compared to patients with high BMI/low WC. These findings suggest that the long-term prognosis of patients after coronary artery revascularization was favored by the baseline high BMI/low WC profile, possibly explaining the obesity paradox and suggesting that assessment of central obesity (either by evaluation of body composition or simply by measurement of WC) should be performed in patients prior to revascularization interventions for a more accurate prognosis [44]. Furthermore, another study demonstrated that patients admitted to hospital for management of acute myocardial infarction had better prognosis when they were overweight or obese, followed by those with normal weight, while underweight patients exhibited the worst prognosis. The authors of that study reported that most of the underweight patients were older with more comorbidities compared to overweight patients, offering a probable explanation for this paradoxical finding [40].

In our study, among patients with myocardial ischemia, female patients had a significantly higher BMI compared to males. In the subgroup analysis within each sex, there was significantly increased BMI (*p* = 0.002) and WC (*p* = 0.020) in women with myocardial ischemia compared to those without ischemia. Such a difference was not found within the male patient subgroup (Figure 1 and Figure 2). In accordance with our findings, Hammer et al. reported a correlation of CAD with BMI and WC in women [45]. Another study in Chinese females reported that a high BMI in addition to hypertension accelerated the risk for type II diabetes mellitus [46], an additional risk for cardiovascular events. Similarly, Chen et al. studied Chinese people with normal BMIs for their predisposition to develop type II diabetes mellitus depending on their WC. Interestingly, they reported that prediabetic patients, especially women, with increased WC had a higher risk to develop type II diabetes mellitus [38].

Although in our study, the univariate analysis showed that the male sex had the strongest association with myocardial ischemia followed by a history of known CAD, in multivariable logistic regression, male sex and BMI were found to be the only two independent predictors of myocardial ischemia. Another study by Luan et al., studied 645 patients (213 females and 431 males) with normal BMI, and reported that within the group of patients with increased WC, men exhibited a higher incidence of CAD than women, after adjusting for other cardiovascular risk factors [47].

The limitations of our study included its retrospective nature and a rather limited population size. Although the patients were consecutive unselected patients, the findings may not be applicable to the general population. Moreover, there was not a detailed analysis of body composition (BMI and WC give only partial information), and no information about inflammatory parameters was given. Finally, the lack of assessment of adipose tissue dysfunction is another limitation of the present study.

## 5. Conclusions

Our study in an unselected population undergoing MPI SPECT showed increased incidence of myocardial ischemia in men compared to women. In subgroup analysis within the ischemia group, BMI was significantly higher in females compared to males. Multivariable logistic regression showed that male sex and BMI were the only two independent predictors of ischemia in our patient population. Our findings suggest that by reducing BMI, cardiovascular risk may be reduced. Whether obesity increases cardiovascular risk per se or indirectly via hyperlipidemia, hypertension, and type II diabetes, conditions that frequently co-exist with obesity, needs more research [48,49].

## Figures and Tables

**Figure 1 medicina-58-00987-f001:**
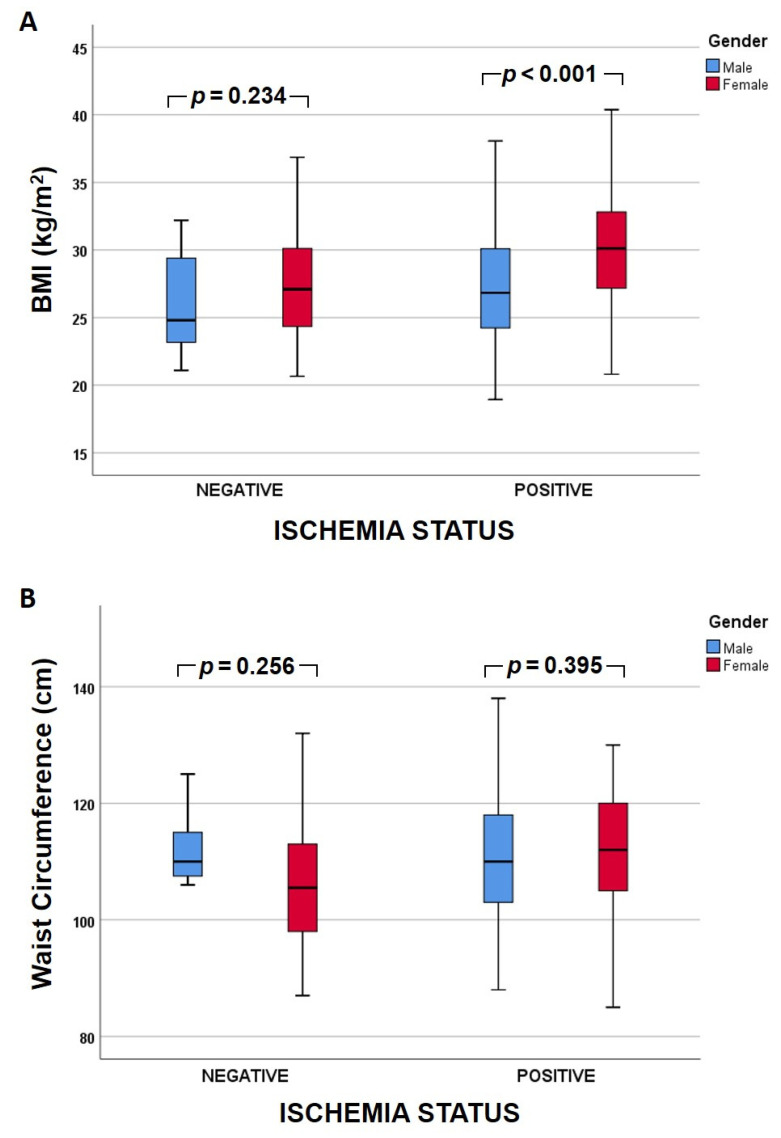
Clustered box plot of (**A**) body mass index (BMI) and (**Β**) waist circumference between males and females according to ischemia status (primary grouping). Each box plot shows the median (bold line), the 25th and 75th quartile (lower and upper box border, respectively), and the minimum and maximum values (whiskers).

**Figure 2 medicina-58-00987-f002:**
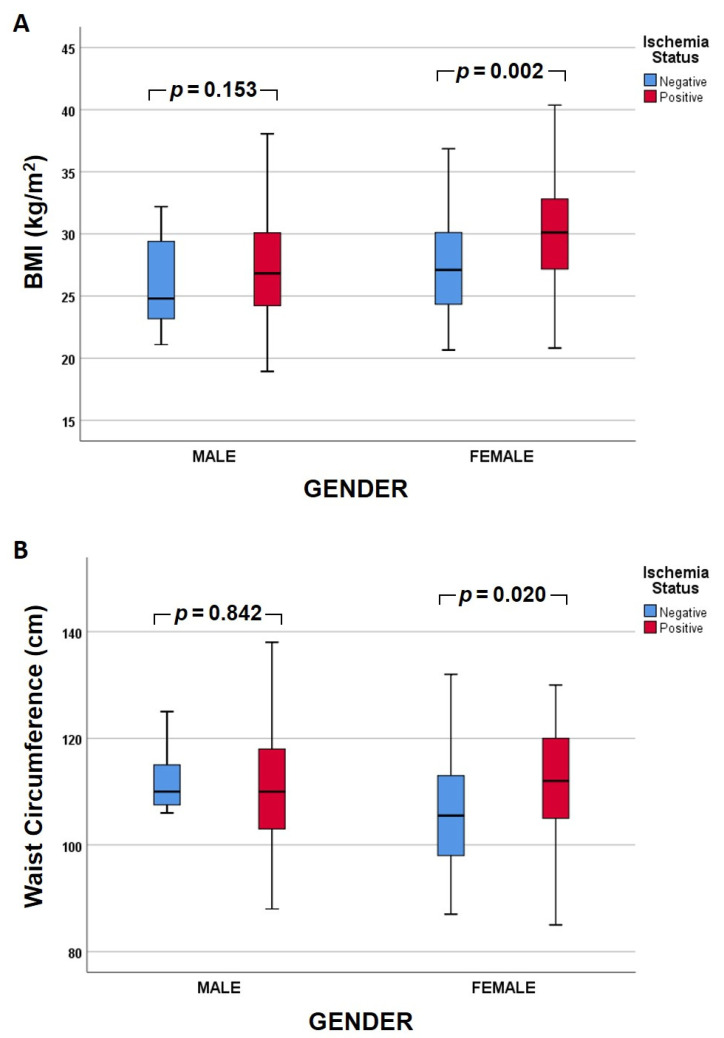
Clustered box plot of (**A**) body mass index (BMI)and (**Β**) waist circumference between the ischemia and non-ischemia groups according to sex (primary grouping). Each box plot shows the median (bold line), the 25th and 75th quartile (lower and upper box border, respectively), and the minimum and maximum values (whiskers).

**Table 1 medicina-58-00987-t001:** Baseline characteristics according to the absence or presence of ischemia in myocardial scintigraphy.

Characteristic	Non-Ischemia (*n* = 69)		Ischemia (*n* = 239)		*p*-Value
Age (years)	70 (65–78)		69 (60–76)		0.136
Age > 65 years	46	(66.7%)	137	(57.6%)	0.175
Female sex	50	(72.5%)	67	(28%)	<0.001
BMI (kg/m^2^)	26.22 (23.77–30.07)		27.76 (24.49–30.86)		0.069
Overweight/Obese	43	(62.3%)	168	(70.3%)	0.209
Waist circumference (cm)	108 (100–114)		110 (103–119)		0.039
Diabetes	17	(24.6%)	83	(34.7%)	0.115
Hypertension	53	(76.8%)	182	(76.2%)	0.909
Hypercholesterolemia	51	(73.9%)	190	(79.5%)	0.322
Smoking	13	(18.8%)	63	(26.4%)	0.202
Family history CAD	31	(44.9%)	85	(35.6%)	0.157
History of CAD	14	(20.3%)	110	(46%)	<0.001
Previous MI	8	(11.6%)	66	(27.6%)	0.006
Previous PCI	10	(14.5%)	73	(30.5%)	0.008
Previous CABG	3	(4.3%)	22	(9.2%)	0.193

Continuous variables are presented as medians (IQR); categorical variables as counts (%). BMI body mass index, CABG coronary artery bypass graft, CAD coronary artery disease, MI myocardial infarction, PCI percutaneous coronary intervention.

**Table 2 medicina-58-00987-t002:** Univariate analysis regarding the association of parameters with the presence of ischemia in myocardial scintigraphy.

Predictor	Odds Ratio	95% CI	*p*-Value
Age (per 1 year increase)	0.982	0.96–1.01	0.200
BMI (per 1 kg/m^2^ increase)	1.064	0.997–1.14	0.062
Waist circumference (per 1 cm increase)	1.027	1.003–1.05	0.028
Sex (male vs. female)	6.76	3.71–12.3	<0.001
Overweight/Obese	1.431	0.82–2.51	0.210
Diabetes	1.63	0.89–2.99	0.117
Hypertension	0.929	0.50–1.75	0.819
Hypercholesterolemia	1.369	0.73–2.55	0.323
Smoking	1.592	0.82–3.1	0.171
Family history of CAD	0.677	0.393–1.16	0.159
History of CAD	3.35	1.77–6.35	<0.001
Previous MI	2.91	1.32–6.41	0.008
Previous PCI	2.595	1.26–5.36	0.010
Previous CABG	2.23	0.65–7.69	0.204

BMI body mass index, CABG coronary artery bypass graft, CAD coronary artery disease, CI confidence interval, MI myocardial infarction, PCI percutaneous coronary intervention.

**Table 3 medicina-58-00987-t003:** Independent predictors of the presence of ischemia in myocardial scintigraphy.

Predictor	Odds Ratio	95% CI	*p*-Value
Sex (male vs. female)	8.86	4.66–16.83	<0.001
BMI (per 1 kg/m^2^ increase)	1.13	1.05–1.22	0.001

Variables with *p* < 0.2 in univariate analyses added in the multivariable model: age, sex, body mass index, waist circumference, diabetes, smoking, family history of coronary artery disease, history of coronary artery disease. BMI: body mass index, CI: confidence interval.

## Data Availability

The data that support the findings of this study are available from the corresponding author upon reasonable request.

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
