# Peer review of "Body Mass Index Is Independently Associated with the Presence of Ischemia in Myocardial Perfusion Imaging"

_medicina, 2022, doi:10.3390/medicina58080987_

Round 1

Reviewer 1 Report

Obesity has been associated with various cardiovascular risk factors, increased incidence of coronary artery disease and myocardial perfusion defects. Authors were invetigated if body mass index (BMI) and waist circumference (WC) were associated with myocardial perfusion defects using myocardial perfusion imaging (MPI) with single photon emission computed tomography (SPECT).

Authors found increase incidence of myocardial ischemia in men compared to women and BMI was significantly higher in females compared to males. Authors also concluded that BMI was an independent predictor of ischemia, suggesting that measures to reduce it could decrease the cardiovascular risk. The authors summarized and discussed the data well. 

Author Response

"Please see the attachement"

Reviewer 2 Report

Sioka et al. prepared a research article, in which it was established that male gender and BMI are independent predictors of myocardial ischemia in MPI SPECT in multivariate analysis but there is no significant difference between BMI in “ischemic” and “non-ischemic” group in univariate analysis. It should be said that generally, all these findings are currently just commonly known because obesity increases cardiovascular risk according to different mechanisms, so the novelty of this paper is low. An interesting aspect of this paper, which increases its attractiveness, is the fact, that cardiological assessment was performed not by invasive angiography or computed tomography-based angiography of coronary arteries, but by nuclear medicine. Although generally background and findings of this paper are of low novelty, the scientific character of this paper is maintained by a well-described methodology, satisfactory statistical analysis, clear and scientific way of presenting results, and generally well-conducted discussion. So taking into account mentioned above aspects, this paper may be considered for publication in Medicina, but some aspects of the manuscript should be significantly improved.

Major revisions:

1) The introduction is too short and too laconic, in my opinion. Although the Authors have described the most important aspects of myocardial ischemia and the diagnostic value of nuclear medicine methods in a satisfactory way, very little was said about obesity. Although obesity increases the risk of cardiometabolic disorders, different phenotypes of obesity should be elucidated, such as metabolically healthy obesity and metabolically unhealthy obesity. Moreover, it is worth mentioning that obesity is associated with increased oxidative stress, apart from a phenotype of obesity. On the other hand, the role of oxidative stress in the pathogenesis of cardiovascular diseases is well established. When we talk about metabolic syndrome, which often occurs in the course of obesity, “obesity and insulin resistance” is the component of metabolic syndrome most strongly associated with oxidative stress. Some molecular mechanisms linking obesity and the pathogenesis of cardiovascular diseases should be mentioned. In my opinion, the introduction should be based on more references. The mentioned above information may be found, for example, in the following high-quality and up-to-date references: 10.1155/2021/9987352, 10.3390/antiox11050856, 10.3390/antiox11010079, 10.3390/ijms22157797, 10.3390/ijms22094798, 10.1016/j.metabol.2021.154766, 10.3390/cells10030629, 10.3390/ijerph17041320.

2) Explain more precisely, please, why six persons were excluded. What information was lacking?

3) In lines 62/63 there is the following text: “who fulfilled the study inclusion criteria”. But inclusion criteria were not described. I have understood that all patients who underwent MPI in 2016 were considered to participate in the study, and subsequently, six patients were excluded due to a lack of information on medical history. So, if I well understood, there cannot be told about inclusion criteria. Moreover, in the discussion, the Authors wrote about “unselected patients”, which suggests that there are no specific inclusion criteria.

4) In lines 102/103 the Authors wrote: “BMI was borderline higher without reaching statistical significance (p=0.069)”. In my opinion, it is incorrectly told from the statistical point of view. It may be eventually told that “mean BMI was borderline higher without reaching statistical significance (p=0.069)”, or better “no statistically significant difference in BMI between patients with and without myocardial ischemia was found in univariate analysis”.

5) Please, explain why “variables with p<0.2 in univariate analyses added in the multivariable model” (explain why exactly 0,2).

6) All limitations of the study should be described. I think about for example lack of analysis of body composition (BMI and WC give only partial information), lack of information about inflammatory parameters, and lack of assessment of adipose tissue dysfunction. I do not agree with the sentence: “In addition, the presence of myocardial ischemia on MPI SPECT may not necessarily indicate the presence of angiographic CAD”. In my opinion, it is not a limitation and there is no contradiction between diagnosed CAD and lack of diminished myocardial perfusion, and on the other hand, there is no contradiction between lack of lesions in angiography and the presence of myocardial ischemia. Myocardial ischemia may be caused by the pathology of microvessels, not assessed in angiography. On the other hand, if a patient is after successful revascularization, and is well pharmacologically treated, the patient has further diagnosed CAD, but maybe there are no myocardial perfusion disturbances because currently there are no significant changes in coronary arteries.

7) Who was responsible for the measurement of height, body mass, and waist circumference? I want to be sure, that the measurements are reliable, accurate, and repeatable.

Minor revisions:

1) Minor editorial corrections should be introduced in the text, for example, in line 14 there is “hadmyocardial” whereas it should be “ had myocardial”. A similar situation applies to lines: 13, 15, 16, 77 (I noticed only in mentioned lines, maybe there are also in another).

2) In line no. 46 there is “acute coronary events”. In my opinion, it should be “acute coronary syndromes”. Similarly in lines 47/48, it should be not “acute coronary episodes”, but “acute coronary syndromes”.

3) In my opinion, it should be “median (IQR) age, …” in line 99.

4) All used abbreviations should be explained in the table at the end of the main text.

5) The list of references should be prepared in accordance with the MDPI rules.

6) English should be checked by a language specialist.

Author Response

"Please see the attachement"

Round 2

Reviewer 2 Report

I received for review a revised version of the manuscript prepared by Sioka et al. entitled: "Body mass index is independently associated with the presence of ischemia in myocardial perfusion imaging".

In my opinion, P. T. The Authors of the publication responded well to the comments contained in the review and improved the manuscript in a satisfactory manner. I recommend the paper for publication in its current form.

Thank you for inviting me to the review. I congratulate the authors and wish them success in their further scientific work.